# Kinin B2 Receptor Activation Prevents the Evolution of Alzheimer’s Disease Pathological Characteristics in a Transgenic Mouse Model

**DOI:** 10.3390/ph13100288

**Published:** 2020-10-01

**Authors:** Marielza Andrade Nunes, Mariana Toricelli, Natalia Mendes Schöwe, Helena Nascimento Malerba, Karis Ester Dong-Creste, Daniela Moura Azevedo Tuma Farah, Katia De Angelis, Maria Claudia Irigoyen, Fernand Gobeil, Tânia Araujo Viel, Hudson Sousa Buck

**Affiliations:** 1Department of Physiological Sciences, Santa Casa de Sao Paulo School of Medical Sciences, Sao Paulo 01221-020, Brazil; marielza@ortobioclinica.com.br (M.A.N.); maritoricelli@gmail.com (M.T.); karisdong@gmail.com (K.E.D.-C.); 2School of Arts, Sciences and Humanities, University of Sao Paulo, Sao Paulo 03828-080, Brazil; na_schowe@hotmail.com (N.M.S.); hnmalerba@gmail.com (H.N.M.); taniaviel@usp.br (T.A.V.); 3Heart Institute (Incor), Hypertension Unit, University of Sao Paulo, Sao Paulo 05403-900, Brazil; danielafarah@hotmail.com (D.M.A.T.F.); hipirigoyen@incor.usp.br (M.C.I.); 4Department of Physiology, Federal University of São Paulo (UNIFESP), Sao Paulo 04023-901, Brazil; prof.kangelis@yahoo.com.br; 5Translational Physiology Laboratory, Universidade Nove de Julho (UNINOVE), Sao Paulo 01504-001, Brazil; 6Department of Pharmacology and Physiology, Université de Sherbrooke, Sherbrooke, QC J1H 5N4, Canada; Fernand.Junior.Gobeil@USherbrooke.ca

**Keywords:** Alzheimer’s disease, bradykinin, neurodegenerative diseases, B2 receptor, kallikrein–kinin system

## Abstract

Background: Alzheimer’s disease is mainly characterized by remarkable neurodegeneration in brain areas related to memory formation. This progressive neurodegeneration causes cognitive impairment, changes in behavior, functional disability, and even death. Our group has demonstrated changes in the kallikrein–kinin system (KKS) in Alzheimer’s disease (AD) experimental models, but there is a lack of evidence about the role of the KKS in Alzheimer’s disease. Aim: In order to answer this question, we evaluated the potential of the kinin B2 receptors (BKB2R) to modify AD characteristics, particularly memory impairment, neurodegeneration, and Aβ peptide deposition. Methods: To assess the effects of B2, we used transgenic Alzheimer’s disease mice treated with B2 receptor (B2R) agonists and antagonists, and performed behavioral and biochemical tests. In addition, we performed organotypic hippocampal culture of wild-type (WT) and transgenic (TG) animals, where the density of cytokines, neurotrophin BDNF, activated astrocyte marker S100B, and cell death were analyzed after treatments. Results: Treatment with the B2R agonist preserved the spatial memory of transgenic mice and decreased amyloid plaque deposition. In organotypic hippocampal culture, treatment with B2R agonist decreased cell death, neuroinflammation, and S100B levels, and increased BDNF release. Conclusions: Our results indicate that the kallikrein–kinin system plays a beneficial role in Alzheimer’s disease through B2R activation. The use of B2R agonists could, therefore, be a possible therapeutic option for patients diagnosed with Alzheimer’s disease.

## 1. Introduction

Aging is a natural process that results in several biochemical, morphological, and physiological changes. The world’s population is aging, with changes in lifestyle resulting in a greater number of older people, and, consequently, an increasing incidence of neurodegenerative diseases. Among them, Alzheimer’s disease (AD) has the highest prevalence and is characterized by progressive neurodegeneration, cognitive deficits, behavioral alterations, and severe functional incapacity [1]. No disease modifying therapy has yet been found for the treatment of AD patients, although our group has previously shown that microdose lithium treatment can prevent and stabilize AD-related symptoms [2] but without any regression of the disease. Other research from our group has shown the involvement of kinin receptors in aging and Alzheimer’s disease [3,4,5]. Kinins act on two receptors, B1 and B2, coupled to different G proteins. These receptors have only 36% homology to each other in their amino acid sequences, but their signaling pathways are very similar. Previous reports from our group showed an increase in both kinin B1 and B2 receptor (BKB1R and BKB2R, respectively) densities after chronic infusion of Aβ peptide, followed by memory impairment in rats [6]. In addition, chronic infusion of Aβ peptide in BKB2R knockout mice leads to premature memory impairment when compared to wild-type animals, as well as an increased number of Aβ plaques. However, memory impairment was absent in BKB1R knockout mice infused with Aβ peptide, with an increase in BKB2R density in memory processing-related areas, suggesting a neuroprotective role for BKB2R [3,4,5]. In recent years, several studies suggested the inhibition or activation of B1 and B2 receptors as therapeutic targets for different brain disorders such as ischemic stroke, temporal lobe epilepsy, febrile illness, spinal cord injury, traumatic brain injury, and in Alzheimer’s disease with controversial results [7]. However, in our knowledge, there are only two kallikrein–kinin system (KKS) modulating drugs approved for commercial use, the B2 receptor antagonist HOE140 (Icatibant) and ecallantide, which are used to treat hereditary angioedema [8,9]. Chronic infusion of BKB1R antagonist in transgenic mice expressing human amyloid precursor protein (APP) abolished amyloidosis, cerebrovascular arrest, and memory deficits [10]. It should be noted that the prolonged blockade of BKB1R can lead BK to act preferentially on B2R, which is considered neuroprotective, and capable of avoiding memory loss [11]. These data point to BKB2R activation as a potential therapeutic target in AD. Moreover, another study by our group showed that the B2R only played a neuroprotective and anti-inflammatory role in organotypic hippocampal slices after BK stimulation in middle-aged adult mice (12 months old), but not in samples from young adult animals (6 months), thus showing its age-modulated neuroprotective role [12]. However, the role of the BKB2R in Alzheimer’s disease has not yet been fully elucidated. Our goal was, therefore, to evaluate the potential of BKB2R in modifying AD characteristics, mainly memory impairment, neurodegeneration, and Aβ plaques deposition. After treatment with the B2R agonist, TG animals had their memory preserved compared with the control group. The B2R agonist was able to protect cells from degeneration, decrease the amount of amyloid plaques, decrease pro-inflammatory cytokines levels, and increase BDNF levels. Our results show that BK, via its BKB2R, has a neuroprotective role in transgenic Alzheimer’s disease animals. Thus, the use of B2R agonists could, therefore, be a possible therapeutic option for patients diagnosed with Alzheimer’s disease.

## 2. Results

### 2.1. Kinin B2R Agonist Preserves Spatial Memory of Transgenic Alzheimer’s Disease Mice

To analyze the spatial memory and learning of treated and untreated animals, we chose the Barnes maze test. Briefly, the test measures the animals’ ability to learn and remember the target location, thus being very useful for assessing the effect of different drugs on cognition. To evaluate the effect of the biostable B2R-selective agonist NG291 and the B2R antagonist bradyzide (BDZ) on Alzheimer’s disease development, 12 month-old C57Bl/6 non-transgenic littermate and B6.Cg-Tg (PDGFB-APPSwInd) transgenic mice (WT and TG, respectively) were treated for 60 days (subcutaneous infusion). Control animals were treated with artificial cerebral spinal fluid (CSF), which was the vehicle used to dilute all the peptides. Before the treatment began, all the animals were exposed for five consecutive days to the Barnes maze for learning the task; memory was evaluated seven days later (day 12) and no memory impairments were observed (data not shown). After 60 days of treatment, on day 72, spatial memory was assessed again. Transgenic mice treated with CSF or BDZ showed significant worsening of spatial memory (4.2-fold, *p* < 0.05 and 4.8-fold, *p* < 0.05, respectively) (Figure 1A). On the other hand, transgenic mice treated with NG291 showed no memory impairment, indicating that the B2R agonist infusion protected against the loss of spatial memory (Figure 1A). In addition, on day 72, TG animals treated with NG291 showed a significant decrease (4.4-fold, *p* < 0.05) in the running distance to reach the target compared to day 12 (Figure 1B). In the same test, mice treated with CSF or BDZ showed significant increase in the running distance (2.2-fold, *p* < 0.001 and 3.5-fold, *p* < 0.05, respectively) (Figure 1B). With respect to the WT mice, no memory disruption was observed, regardless of the treatment (Figure 1A).

### 2.2. B2R Does Not Alter Motor Activity but Plays a Role in Animal Sensitivity

The activity of animals infused with vehicle or kinin B2R analogues was evaluated in a motor activity test in order to eliminate the possibility that any poor performance in the tests was due to motor deficit. No change in ambulation was observed in treated WT and TG mice compared to WT and TG mice infused with CSF (Figure 2A). In the same way, no change in rearing behavior was observed (Figure 2B).

Due to the action of kinin analogs on pain stimulus, we also evaluated animal nociception. Paw withdrawal threshold was measured using the electronic von Frey test. The test was performed at the beginning of the protocol, after the motor activity test and before the treatment, to obtain baseline values and at the end of the protocol to assess possible hyperalgesia or hypoalgesia. The von Frey filaments are used to evaluate tissue sensitivity to mechanical stimulation. We observed that TG mice treated with the B2R agonist NG291 showed a significant reduction in pressure sensitivity (mechanosensitivity) compared with CSF- and BDZ-treated mice (7.0-fold, *p* < 0.05 and 9.0-fold, *p* < 0.01, respectively) (Figure 2C). Wild type mice treated with B2R antagonist BDZ and TG mice treated with CSF showed increased mechanosensitivity compared with CSF-infused mice (5.6-fold, *p* < 0.01 and 4.4-fold, *p* < 0.05, respectively) (Figure 2C).

### 2.3. B2R Agonist Increases Cerebral Blood Flow of WT and TG Animals

To evaluate the effects of the B2R agonist on cerebral blood flow, 12-month-old WT and TG mice were infused for 14 days with NG291 through an osmotic minipump. As the B2R antagonist had shown no protective effects in previous experiments, infusion with this peptide was not performed in order to avoid unnecessary use of animals. Possible changes in cerebral blood flow due to B2R agonist action were evaluated by infusion of yellow microspheres and subsequent quantification of the microspheres in the following target organs: cerebral hemispheres (right and left); frontal cortex; heart; and left and right kidneys [13]. This quantification was performed using a spectrophotometer (λ = 465 nm).

WT and TG animals treated with B2R agonist showed a significant increase in cerebral hemisphere blood flow (2.68 ± 0.79 mL/min/g tissue, *n* = 7, and 1.48 ± 0.33 mL/min/g tissue, *n* = 6, *p* < 0.05, respectively) compared to untreated animals (1.76 ± 0.51 mL/min/g tissue, *n* = 6 and 1.00 ± 0.25 mL/min/g tissue, *n* = 4, respectively), but not in the other analyzed organs or region (Figure 3).

### 2.4. Kinin B2R Agonist Decreased Amyloid Plaques in TG Animals

Amyloid plaque density was assessed by histochemical staining with thioflavin-S (Figure 4A). Following treatment with NG291, there was a significant decrease in plaque numbers in both the hippocampus and cortex of TG animals (Figure 4B) when compared to CSF-treated mice, demonstrating that the B2R can exert a neuroprotective function in relation to senile plaque formation. In addition, the blockade of B2R by the treatment with BDZ led to an increase in plaque numbers in the hippocampus.

### 2.5. Treatment with Different KKS Modulators Increased Cell Viability 

In order to evaluate the secreted substances associated with the activation of the BKB2R, we adopted the organotypic hippocampal culture (OHC) model using a reduced number of animals (four per group/treatment). Our laboratory has developed a method that allows the maintenance of healthy hippocampal slices of adult animals (12 months old) for a long period of time [12]. To confirm the viability and good maintenance of the hippocampal slices from the WT and TG animals over the 28 days in vitro (28DIV), we stained the slices with propidium iodide to quantify cell death. Figure 5A–C show that after four weeks, more than 60% of the cells from different areas of the hippocampus remained viable in both the WT and TG animals.

After analyzing the viability, we assessed the effect of different treatments on the cell viability of the OHC from 12-month-old transgenic mice. The hippocampal slices were treated with BK, B1R antagonist, B2R antagonist, and BK plus B1R antagonist. In this last combination, the B1R antagonist was added considering the possible activity of tissue carboxypeptidase M-like enzymes converting BK into the BKB1R agonist desArg9-BK [14]. Kinin B2R antagonist was used to block the effect of endogenous BK. Treatment with the B1R antagonist alone or associated with BK decreased cell death only in the Dentate Gyrus (DG), when compared to the untreated control (0.48- and 0.67-fold, *p* < 0.0001, respectively, Figure 5D). Furthermore, the use of the B2R antagonist was not able to protect against cell death in the DG and CA1 and CA2 areas, but it increased the levels of death in the CA3 area (2.2-fold, *p* < 0.0001) (Figure 5F) suggesting that endogenous BK could have a protective effect. BK was able to protect slices from death in the CA3 region, showing a decrease of 0.64-fold (*p* < 0.0001) (Figure 5F), thus reinforcing the data obtained with the in vivo treatment of AD transgenic mice with the B2R agonist.

### 2.6. B2R Activation Decreases the Release of Inflammatory Markers

One of the main KKS roles discussed in the literature is its effect on inflammation. To better understand the participation of B2R in the release of inflammatory markers in Alzheimer’s disease, we treated the hippocampal slices with different kinin analogues.

In OHC of 12-months-old transgenic mice, treatment with bradykinin significantly decreased IL-6 and CCL4 levels, two important inflammatory mediators (Figure 6B,D). Treatment with the B1R antagonist decreased the release of IL-1a, IL-6, and CCL4 (Figure 6A,B,D).

The B2R antagonist increased IL-1a levels (1.59-fold, *p* < 0.001) (Figure 6A). These data show that BK acting only on the B2R led to a drastic decrease in all evaluated cytokines.

### 2.7. B2R Protected Hippocampal Slices of Transgenic Animals from Brain Damage

To better elucidate possible BKB2R neuroprotective activity, we analyzed the S100b and the neurotrophin BDNF levels after treatments. In 12-month-old TGOHC, BKB1R inhibition was able to decrease S100b levels, whereas B2R inhibition increased S100b release, showing once again that depending on the context, the B2R can act as a neuroprotector (Figure 7A).

Regarding BDNF, OHC of 12-month-old transgenic mice showed lower levels of this neurotrophin (1.24-fold, *p* < 0.05), indicating that the disease itself already promotes a decrease in BDNF levels (Figure 7B). After bradykinin administration, BDNF levels significantly increased (2.19-fold, *p* < 0.001). The same effect was observed after treatment of BK plus the B1R antagonist (1.75-fold, *p* < 0.05) (Figure 7C). Taking our data together, they demonstrate the crucial role of the BKB2R in neuroprotection.

## 3. Discussion

Our research group has been studying the role of the kallikrein–kinin system in Alzheimer’s disease for a number of years. We have shown that 18-month-old mice lacking B1R had better memory following aversive stimulation than same-age wild-type animals; we also found that 12-month-old mice lacking the B2R presented memory deficit. These results suggest that the absence of the B1R decreases cognitive deficit during aging, while the absence of the B2R promotes increased cognitive deficit [3].

In another study, we observed that 3-month-old mice lacking B2R submitted to chronic infusion of the amyloid-β peptide 1-40 (Aβ 1-40) presented severe cognitive loss, while mice lacking B1R showed no memory deficits [4]. We also found increased densities of synapses and BKB2R in KOB1R animals infused with Aβ 1-40. Mice lacking B2R submitted to the same protocol, showed an increase in the number of amyloid plaques compared to the control [5], reinforcing the hypothesis that the B2R has a neuroprotective role. Moreover, we also demonstrated that the chronic infusion of amyloid-β peptide led to an increase in the density of B1R and B2R in brain regions associated with cognitive processes, as well as an increase in BK concentration in cerebrospinal fluid, suggesting the involvement of KKS activation in AD [6,15].

Based on these results, in the present study, we evaluated the potential of BKB2R activation to modify AD properties in a transgenic mouse model. Motor activity was measured to ensure that the results of the memory tests were not affected by changes in these parameters, and no differences were observed among the groups. 

In respect of kinin action on nociception mechanisms, pain sensitivity was evaluated before and after the treatment. The mechanical sensitivity threshold increased in TG mice compared to vehicle-infused WT animals, and treatment with the B2R agonist prevented any increase in sensitivity. Despite the fact that kinin peripheral action on nociception is well established [16], a dose-dependent antinociceptive effect induced by BK stimulation of brain BKB2R was shown [17]. The antinociceptive effect observed after B2R agonist NG291 treatment could, therefore, be due to the activation of central BKB2R, reinforcing the central action of the B2R agonist. In TG mice, infusion of the BKB2R agonist preserved acquired memory, while TG mice treated with vehicle and B2R antagonist showed no preservation of the acquired memory. In the present study, we did not perform treatment using the B1R antagonist as this has already been done previously [10]. In that study, using the same transgenic mice model, the authors observed a similar neuroprotective effect by blocking the BKB1R. As shown in the data from the transgenic mice organotypic hippocampal culture model, there appears to be a synergism between the stimulation of BKB2R, and the antagonism of BKB1R. Importantly, in the present study, and also in a study by Lacoste et al. (2013), the kinin analogues were administered subcutaneously, being able to reach the central nervous system [10].

Amyloid β accumulation, an early event in AD, occurs together with early cerebral vascular dysfunction, inducing disruptions in the normal regulation of cerebral blood flow [18,19]. Hypoperfusion of the right inferior parietal cortex, right middle frontal cortex, and posterior cingulate are associated with dementia [20,21]. The hypoperfusion could lead to hypoxia, which is associated with Tau protein hyperphosphorylation and increased neuronal death [22]. 

BK is a potent vasodilator, participating in the control of cerebral blood flow [23]. Our data showed an increase in cerebral blood flow in the whole hemisphere of WT and TG mice after the 14-day treatment with B2R agonist. This increase in cerebral blood flow could have, at least in part, contributed to memory preservation and to the reduction in amyloid plaque deposition, one of the hallmarks of Alzheimer’s disease [24]. It has already been suggested that a compensatory hemodynamic mechanism could provide protection against pathology in the early stages of AD [25]. 

The decrease in Aβ accumulation in NG291-treated transgenic mice is another important observation. The neurodegeneration observed during the progression of Alzheimer’s disease involves several mechanisms. Some of them are complex and not yet fully understood, but the Aβ peptide has been shown to have an effect in two well-described mechanisms: (1) oxidative stress, which involves mitochondrial dysfunction, activation of NMDA receptors, Ca^2+^-induced excitotoxicity, and the production of reactive oxygen species [26,27,28]; and (2) inflammatory processes, involving enzyme activation, glial activation, and expression of inflammatory proteins and cytokines, and other substances [29,30,31,32]. The transgenic mice used in the present work naturally have a large number of amyloid plaques, which can increase the levels of inflammatory mediator release. In fact, we observed that OHC from 12-month-old transgenic mice not treated with any drugs have high levels of inflammatory cytokines such as IL6, IL1α, CCL2, and CCL4. Interestingly, BK treatment together with the BKB1R antagonist was able to significantly decrease the levels of all the inflammatory mediators analyzed. However, the role of the BKB1R cannot be discarded; Sanden et al. demonstrated that BKB1R is upregulated after lipopolysaccharide (LPS) injection, resulting in an increase in TNFα and IFNɣ [33]. Another study recently published by our group demonstrated the importance of animals’ age with respect to the neuroprotective action of bradykinin and BKB2R [12]. In this study, a decrease in inflammatory markers was observed after treatment with BK only in 12-month-old animals; in 6-month-old animals, kinin receptor activation increased inflammatory markers [12]. Due to technical limitations in obtaining acceptable thinner slices from organotypic hippocampal culture, the present work has the limitation of not showing microglia activation by immunohistochemistry. Other parameters that are part of Alzheimer’s pathophysiology were also evaluated, such as the activation of astrocytes by S100b and the release of the neurotrophin BDNF. The protein S100b is involved in the regulation of several cellular processes and is expressed mainly in astrocytes and released in the extracellular space in response to glutamate, serotonin, inflammatory cytokines, and β-amyloid peptides [34]. S100b has often been used as a biomarker of cellular damage in astrocytes [35]. BDNF is an important neurotrophin that regulates several aspects of neuronal development and function, such as activation of cell survival pathways and neuroplasticity. 

Again, we note the importance of BKB1R antagonism and BKB2R activation by BK in halting the damage caused by beta amyloid deposition. In another study, our group showed that KOB1R animals presented an increase in BKB2R density associated with an increase in synaptophysin [5]. It is known that the intracellular signaling pathway of BKB2R is mediated through the action of protein G, mainly Gq, which stimulates phospholipase C (PLCβ) and phospholipase A2 (PLA2). PLCβ and PLA2 then activate PKC via diacylglycerol (DAG) and increase intracellular Ca^2+^ in the cytoplasm. Through the PLA2 pathway, arachidonic acid is released, which is metabolized by prostaglandin synthase H (PGH) to prostaglandins and reactive oxygen species (ROS), leading to inflammation and oxidative stress [36]. Another activation pathway for this receptor is activation of PLC, converting IP2 into IP3 and DAG, leading to an increase in intracellular Ca^2+^, including the activation of PKC and the release of other neurotransmitters [36].

BKB2R activation can also lead to activation of protein kinase A (PKA) which, concurrently with PKC, can phosphorylate, directly or indirectly, the subunits of NMDA (*N*-methyl-D-aspartic acid) and AMPA (α-amino-3-hydroxy-5-methyl-4-isoxazolepropionic acid) receptors changing the traffic and kinetics of these receptors, thus increasing synaptic efficacy [37]. Parpura et al. showed in mouse astrocyte culture that BK is indirectly involved in the induction of glutamate release. Such release was also observed in rat astrocytes and in Schwann cells [38], by means of an intracellular Ca^2+^ increase mediated by NMDA [39]. Glutamate is involved in synaptic transmission and long-term potentiation (LTP). The signal transduction pathway of BKB2R, through the activation of PKA, PKC, and MAPK (mitogen activating protein kinase) leads to CREB phosphorylation (cAMP Responsive element binding protein), which can directly or indirectly activate glutamate, NMDA, and AMPA receptors [40]. In this way, BKB2R action may be related to the improved spatial memory by increasing the release or action of glutamate. Thus, the increase in BDNF levels, the decrease in S100B, and the absence of cognitive deficit can be explained by the potentiation of this pathway.

Of note, the role of chronic BDNF exposure was observed in glutamatergic neurons in organotypic hippocampal culture. Rauti et al. treated hippocampal slices with BDNF (250 ng/mL) for 72 h and observed increases in spine density and synapse number in pyramidal neurons in the hippocampus [41]. Furthermore, BDNF intracellular overexpression in hippocampal neurons increases both the frequency and the amplitude of the excitatory synapses, therefore indicating the relevance of BDNF [41]. Therefore, in the present study, we demonstrated that the activation of BKB2R by the specific agonists NG291 and BK can become an important tool in the fight against Alzheimer’s disease. Other known drugs can also activate B2R and could be potential tools for the treatment of neurodegenerative diseases, but their therapeutic value, as well as side effects considering the treatment of the elderly, are still to be determined. In this way, the angiotensin-converting enzyme inhibitor (ACEI) can increase the concentration of circulating BK and increase B2R functions as an allosteric enhancer potentiating the actions of BK. In addition, there is the angiotensin II receptor blocker Losartan, which acts as a partial agonist of B2R [42,43,44].

It is important to highlight that the neuroprotective role of NG291 and BK depends on a pre-existing pathological state in which BKB2R plays a leading role in neuroprotection.

## 4. Materials and Methods 

### 4.1. Animals

Twelve-month-old (initial age) transgenic mice (TG) B6.Cg-Tg (PDGFB-APPSwInd) and age-matched non-transgenic (wild-type—WT) littermate mice were used in this study. The animals were provided from our own colony, using breeding males acquired from Jackson Laboratories, USA (Stock Number 006293). According to the Jackson Laboratory, these animals show the complete spectrum of cerebrovascular, cognitive, neuroinflammatory, and amyloid pathologies at 10 months of age. All the animals were genotyped to confirm the presence of the mutant gene according to the protocol described by the Jackson Laboratory. The animals were maintained in littermate groups (three to six) in ventilated cages (food and water ad libitum; room temperature between 24 and 26 °C; humidity at 55%).

All procedures performed were approved in accordance with the Animal Ethics Committee of Santa Casa de São Paulo School of Medical Sciences, under number 001/14.

### 4.2. Animal Treatment

The mice were treated with the biostable BKB2R-selective agonist NG291 [Hyp3,Thi5,NChg7,Thi8]-BK, which exhibits more potent activities than BK in vivo. Unlike BK, it is not susceptible to degradation by ACE (alias kininase II) found in the blood/tissues and, importantly, it does not exhibit activities in vivo (and ex vivo) in BKB2R knockout mice, highlighting its high degree of specificity towards BKB2R [45] synthetized in the laboratory of Dr. Fernand Gobeil; or with the orally active, potent, non-peptide kinin B2R antagonist bradyzide [46,47] (BDZ, Sigma-Aldrich, stock number B1680, Saint Louis, MO, USA). Both analogues were diluted in artificial cerebrospinal fluid (CSF) at a dose of 10 nmol/kg/h. Peptides were infused using a mini-osmotic pump (Alzet, Cupertino, CA, USA, model 1004, 0.11 μL/h) subcutaneously implanted under isoflurane anesthesia. For the treatment period of eight weeks, the animals received a mini-pump that was removed after four weeks and replaced by a new one for the next four weeks. Control animals were treated with vehicle for the same period. The animals were in the following six groups: TGCSF; WTCSF; TGNG291, WTNG291; TGBDZ and WTBDZ.

### 4.3. Motor Activity

Before and after peptide infusion, horizontal and vertical mobility tests were performed using an animal activity cage (model 7430, Ugo Basile, Comerio, Italy) to verify whether there were any mobility deficiencies that could compromise the memory and behavior tasks. The apparatus has vertical sensors that register rearing activity and horizontal sensors that register locomotor activity. The vertical and horizontal movements of the mice were recorded for five minutes as previously described [2].

### 4.4. Spatial Memory

Spatial memory was evaluated using the Barnes maze test, following previously described protocols [2]. This test was developed by Carol Barnes as an alternative to the Morris water maze task for rats and was later adapted for mice. Briefly, the animal was placed in the center of a white, 100 cm diameter board, with 30 holes arranged radially with visual clues on the wall. A dark box, acting as an escape box, was placed under one of the holes. A fluorescent lamp was placed above the center of the arena, and a radio not tuned to a station was played. Animals had 5 min to find the escape box. Once inside the box, the lights and radio were turned off, and the animal was left for 1 min. The learning phase was performed when animals were 12 months old and consisted of five daily sessions. The first memory test was performed seven days later and named day 12. 

The trial was repeated 60 days later, named day 72, when the animals were 14 months old, to assess memory recovery. The time to enter the escape box was recorded. To avoid any bias with respect to the time taken to find the escape box because of the time the animal spends standing still; the time spent in the target quadrant and the distance walked to find the escape box were also measured as complementary parameters.

The trial was recorded using a JVC Everio video camera and analyzed using the “Smart video tracking system” (Panlab, Cornellà, Barcelona, Spain). The evaluated parameters were time to find the target, time spent in the target quadrant (s), and distance (cm) travelled to find the escape box.

### 4.5. Nociception

Nociception was evaluated through paw withdrawal, using the electronic von Frey test. Mice were individually placed in cages with grid floors for 15–30 min in an acoustically insulated room to allow adaptation to the new environment. To elicit the paw-flick response, an increasing force (g) was applied on the plantar surface of the hind using rigid 0.5 mm diameter polypropylene tips (400 g; Model 2391; IITC Life Science). Pain threshold was based on the pressure at which paw withdrawal occurred. This test was done on the 12th day after the Barnes maze test and after the Barnes maze test on the last day (77th day). In each session, the paws were tested three times for hind-paw flexion reflex and the mean value was used. Nociception was measured as the pressure at which paw withdrawal occurred by subtracting the value of session one from the value of session two.

### 4.6. Blood Flow Analysis

At 12 months of age, WT (*n* = 6) and TG mice (*n* = 4) were infused with cerebrospinal fluid (CSF), and WT (*n* = 7) and TG mice (*n* = 6) were infused with BKB2R agonist NG291 for 14 days (10 nmol/kg/h). After this period, mice were anesthetized with xylazine 10–12.5 mg·kg^−1^ and ketamine 80–100 mg·kg^−1^ for implantation of catheters filled with saline into the femoral artery and left ventricle (via the right carotid artery) for infusion of the colored microspheres (CM). The catheters were fixed using suture thread.

A blood flow analysis using microspheres was then performed using the method described in De Angelis et al., 2005 [13]. Yellow Dye-Trak microspheres were sonicated for three minutes before ventricular infusion. Briefly, a spiral of Tygon tubing (40.4 cm) filled with 90 µL of CM was placed between the left ventricular catheter and a 1-mL syringe containing 0.5 mL pre-warmed (40 °C). An infusion pump was used to inject the CM (Infusion Pump 22; Harvard Apparatus, South Natick, MA, USA). Using an infusion and withdrawal pump (Harvard Apparatus) connected to the abdominal aorta catheter, reference blood samples were collected. Reference blood samples were collected from the femoral artery starting 10 s before the CM infusion and was sustained for 75 s. The CM and physiological solution infusion were infused for 50 s (0.18 mL/min). At the end of infusion, the animals were euthanized with an overdose of thiopental and the target organs were collected (heart, kidneys, liver, and brain) to determine regional blood flow. The extraction of the dyes from the isolated dried microspheres was done in the presence of 250 µL of dimethylformamide (Sigma-Aldrich) added to each tube, which was vigorously vortexed for ten seconds. Then, samples were centrifuged (2000 g/10 min), the supernatant was collected, and the absorbance was determined with a spectrophotometer (<1.8-nm slit width; DU 640, Beckmann Instruments, Inc., Fullerton, CA, USA) using a 200-µL quartz cuvette (Sigma-Aldrich, Saint Louis, MO, USA). The peaks of the absorption spectrum for yellow were obtained at 465 nm. The acceptable absorbance was fixed at a minimum of 0.010 absorbance unit.

### 4.7. Quantification of Amyloid-β Plaques

Thioflavin-S was used to quantify amyloid plaque density. Slides were dried for 30 min, and then, washed twice for three minutes with PBS containing 0.1% Triton X-100 (PBST). After that, the slides were immersed in thioflavin-S (Sigma-Aldrich T1892, Darmstadt, Germany) solution (0.1%) diluted in PBST for 5 min, then washed for 3 min in PBST 0.1% and fixed in alcohol 70% for 5 min. Slides were then washed three times for 3 min in PBS 0.1% and coverslipped with fluoroshield with DAPI mounting medium (Sigma-Aldrich, F6057, Darmstadt, Germany) and stored at 4 °C. Plaques in the cortex and hippocampus were manually counted by fluorescence microscopy and quantified as number of plaques per slice using a Leica Inverted microscope (Leica DMi8, Wetzlar, Germany). For each animal, four slices between −2 and −2.5 mm in relation to bregma were evaluated and the mean number of plaques used for the statistical analysis [48].

### 4.8. Preparation of Organotypic Hippocampal Cultures (OHCs)

Organotypic hippocampal culture was prepared according to previous studies from our group [12,49]. Briefly, mice were anesthetized, euthanized by decapitation, their brains were rapidly removed, and 300 µm sagittal slices (6–9 per brain) were taken using a vibratome (Leica VT1000S, Wetzlar, Germany) and dissected to isolate the hippocampus. Four to six slices were placed onto 0.4 mm pore membrane inserts (Millipore PICM03050, Burlington, MA, USA). In the first seven days, a nutrient enriched culture medium was used containing: 50% MEM with Glutamax-1 (Gibco: 42360-032, Gaithersburg, MD, USA), 1 mM GlutaMAXTM Supplement (Gibco: 35050061, Gaithersburg, MD, USA), 10% HAM’S F-10 Nutrient mix (Gibco: 12390035, Gaithersburg, MD, USA), 25% heat-inactivated horse serum (Gibco: 26050-070, Gaithersburg, MD, USA), 1 mM Sodium Pyruvate (Gibco: 1360070, Gaithersburg, MD, USA), 37 mM D-Glucose (Sigma-Aldrich: G8270, Saint Louis, MO, USA), 25% HBSS (Gibco: 14025-092, Gaithersburg, MD, USA), and 1% Antibiotic/Antimycotic Gibco: 15240-062, Gaithersburg, MD, USA). After this period, the inserts were kept in 1 mL of maintenance medium. The OHCs were maintained in incubators at 37 °C, 5% CO_2_ for up to 4 weeks.

### 4.9. OHC Drug Treatment

After 1 week of stabilization, OHCs from 12-month-old (12-OHC) animals were treated with 300pM B2R antagonist (Bradyzide—B2RAnt), 300 pM of BK, 200pM of B1R antagonist (des-Arg10-HOE140—B1RAnt), or 300 pM of BK + 200 pM of B1R antagonist (BK + B1RAnt) added to 1 mL of maintenance culture medium. All drugs were purchased from Sigma-Aldrich (Saint Louis, MO, USA). Every three days, the culture medium containing the treatments was replaced. The B1R antagonist [desArg10]-Hoe140 was added considering the possible activity of tissue carboxypeptidase M-like enzymes converting BK into the BKB1R agonist desArg^9^-BK [14].

### 4.10. Cell Death Evaluation

Cell death evaluation was done using propidium iodide staining, a widely accepted, simple, and reliable assay to evaluate cell death in organotypic hippocampal slice cultures [12,49,50]. For that, hippocampi slices were washed with 1 mL of PBS, sufficient to withdraw the residual culture medium, and incubated with 1 µg/mL of propidium iodide (Sigma-Aldrich, Saint Louis, MO, USA) during 15 min in absence of light. At the end of incubation period, slices were washed with PBS and images were acquired using a ZOE^TM^ Fluorescent Cell Imager (Bio-Rad Laboratories, Hercules, CA, USA). The software ImageJ [51] was used to analyze the labeled proportional area.

### 4.11. Inflammatory Markers Evaluation

Inflammatory markers were evaluated by assessing the levels of MCP-1/JE/CCL2, MIP-1B/CCL4, and interleukin 1α and 6. These markers were measured in the OHCs culture medium using a Magnetic Luminex^®^ Assay—Mouse Premixed Multi-Analyte Kit R&D systems (Minneapolis, MN, USA), according to the manufacturer’s instructions.

Release of BDNF and S100b by OHCs was quantified in the medium by a Quantikine Elisa Total BDNF Immunoassay^®^ (R&D systems, Minneapolis, MN, USA) and by a Mouse S100b (S100 Calcium Binding Protein B) Elisa Kit (Elabscience, Houston, TX, USA), respectively, following the manufacturer’s instructions.

### 4.12. Statistical Analysis

Behavioral data obtained in the Barnes maze test, locomotion, rearing, and mechanosensitivity test were analyzed by one-way ANOVA, followed by Tukey’s multiple comparison tests. Data related to cerebral blood flow and Figure 7B were analyzed by Student’s *t*-test. Data related to amyloid plaques quantification and organotypic hippocampal culture were analyzed with one-way ANOVA followed by Dunnett’s multiple comparison test considering the CSF or the control group (Ctrl), the references for comparisons. All analyses were performed using GraphPad Prism 8.0 (GraphPad Software Inc., San Diego, CA, USA). All data were expressed as means ± SEM. Only probability values (*p*) less than 0.05 were considered statistically significant.

## 5. Conclusions

In this study, the importance of the BKB2R with respect to spatial memory improvement, a reduction in amyloid plaques, and increased blood flow in transgenic Alzheimer’s disease mice was shown. Moreover, in 12-TgOHC, we observed that the modulation of the kallikrein–kinin system was important to decrease cell death and cell damage, inflammatory markers, and increase in BDNF levels. These data allow us to highlight BKB2R and BKB1R and the possible important role they might play in pharmacotherapy for Alzheimer’s disease. In this way, the present study is one step further in the comprehension of the involvement and importance of the kallikrein–kinin system in Alzheimer’s disease and its pharmacological modulation.

## Figures and Tables

**Figure 1 pharmaceuticals-13-00288-f001:**
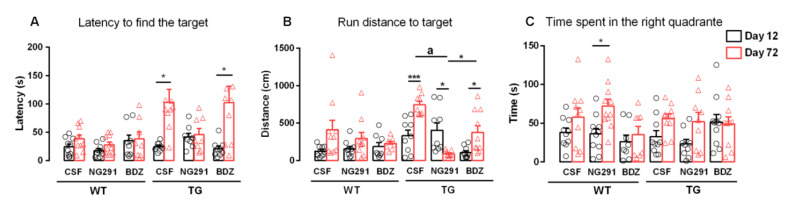
Effect of BK analogues treatment on spatial memory. Spatial memory was evaluated using the Barnes maze test with the time taken to enter the escape box being recorded. Data show that kinin B2R agonist NG291 treatment prevented the memory impairment observed in vehicle and B2R antagonist treated transgenic mice. WT—C57Bl/6 non-transgenic littermate mice; TG—B6.Cg-Tg (PDGFB-APPSwInd) transgenic mice; CSF—Artificial cerebral spinal fluid; NG291—B2R agonist; BDZ—B2R antagonist bradyzide. WTCSF (*n*) = 10, WTNG291 (*n*) = 11, WTBDZ (*n*) = 8, TGCSF (*n*) = 9, TGNG291 (*n*) = 8, TGBDZ (*n*) = 9, Histograms and vertical bars are means ± SEM. Data were analyzed with one-way ANOVA followed by Tukey’s multiple comparison test. * *p* < 0.05; *** *p* < 0.001; a *p* < 0.0001.

**Figure 2 pharmaceuticals-13-00288-f002:**
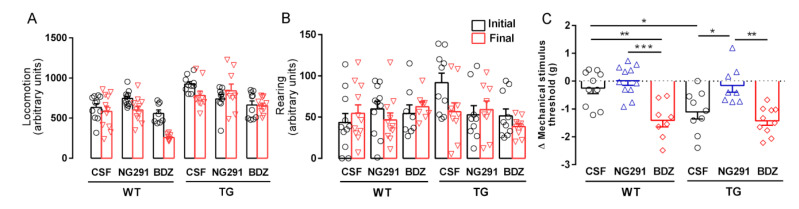
Kallikrein–kinin system does not change motor activity but has an influence on mechanosensitivity of WT and TG mice. Motor activity evaluation of WT and TG mice, submitted or not to treatments. Panel (**A**): locomotion. Panel (**B**): rearing. Panel (**C**): mechanosensitivity; WTCSF (*n*) = 11, WTNG291 (*n*) = 11, WTBDZ (*n*) = 8, TGCSF (*n*) = 10, TGNG291 (*n*) = 9, TGBDZ (*n*) = 10. Histograms and vertical bars are means ± SEM. Data were analyzed with one-way ANOVA followed by Tukey’s multiple comparison test. * *p* < 0.05; ** *p* < 0.01; *** *p* < 0.001.

**Figure 3 pharmaceuticals-13-00288-f003:**
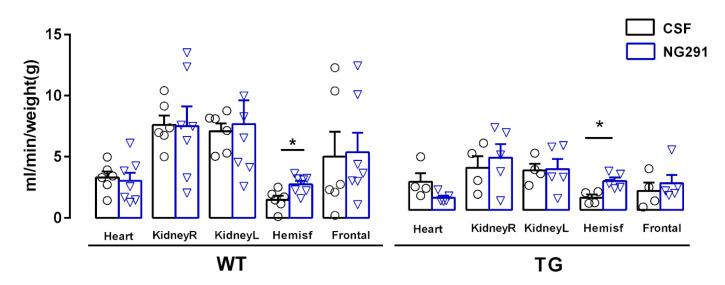
Stimulation of B2R improves cerebral blood flow in WT and TG animals. Blood flow was evaluated in different organs of WT and TG animals treated or not with B2R agonist NG291. Data were analyzed with Student’s *t*-test. * *p* < 0.05.

**Figure 4 pharmaceuticals-13-00288-f004:**
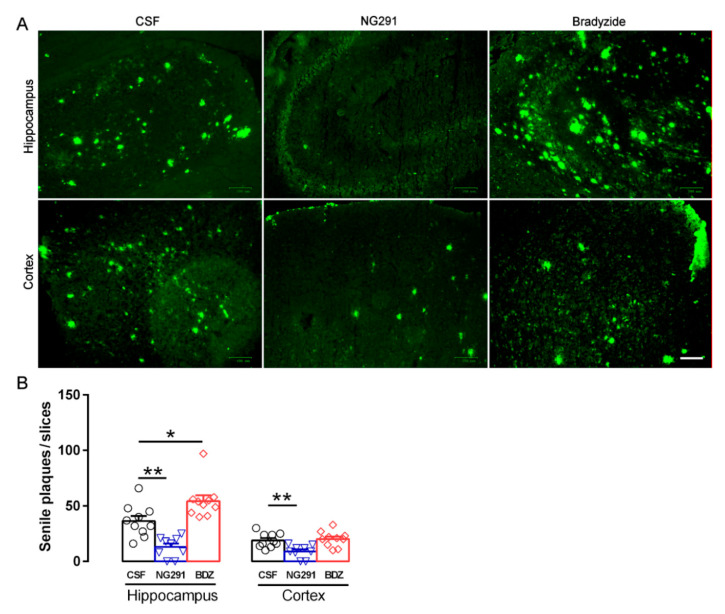
Activation of the B2R decreased the number of Aβ plaques. (**A**) Evaluation of amyloid plaques by the thioflavin-S method. (**B**) Transgenic animals treated with B2R agonist showed less plaque when compared to untreated TG animals. NG291—B2R agonist; BDZ—B2R antagonist. Data were analyzed with one-way ANOVA followed by Dunnett’s multiple comparison test. * *p* < 0.05, ** *p* < 0.01. Scale bar is 100 µm. CSF (*n*) = 10, NG291 (*n*) = 9, BDZ (*n*) = 10.

**Figure 5 pharmaceuticals-13-00288-f005:**
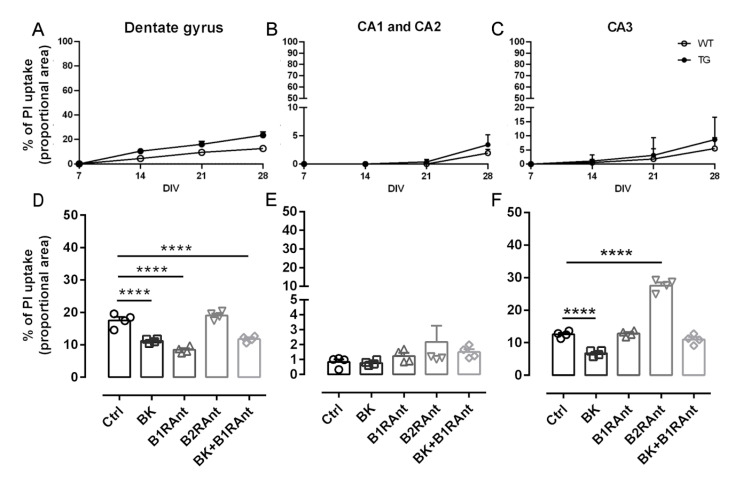
Hippocampal slices of WT and TG animals remained viable over four weeks, and the B2 receptor preserved different areas of the hippocampus of TG animals from cell death. (**A**–**C**) Percentage of cell death along 28 DIV of six hippocampal slices of three 12-month-old wild-type and TG mice. DIV—days in vitro. (**D**–**F**) Percentage of cell death in hippocampal slices of 12-month-old TG animals treated with 300 pM BK, 200 pM B1R antagonist ([des-Arg^10^]-HOE 140), or 300 pM B2R antagonist (Bradyzide) for 20 days (20 DIV). After 20 days, the slices were stained with 1 μg/mL of propidium iodide; images were taken with a Zoe fluorescence microscope and analyzed with ImageJ software. **** *p* < 0.0001. Data were analyzed with one-way ANOVA followed by Dunnett’s multiple comparison test. Technical quadruplicate of biological samples of four animals were used for each treatment.

**Figure 6 pharmaceuticals-13-00288-f006:**
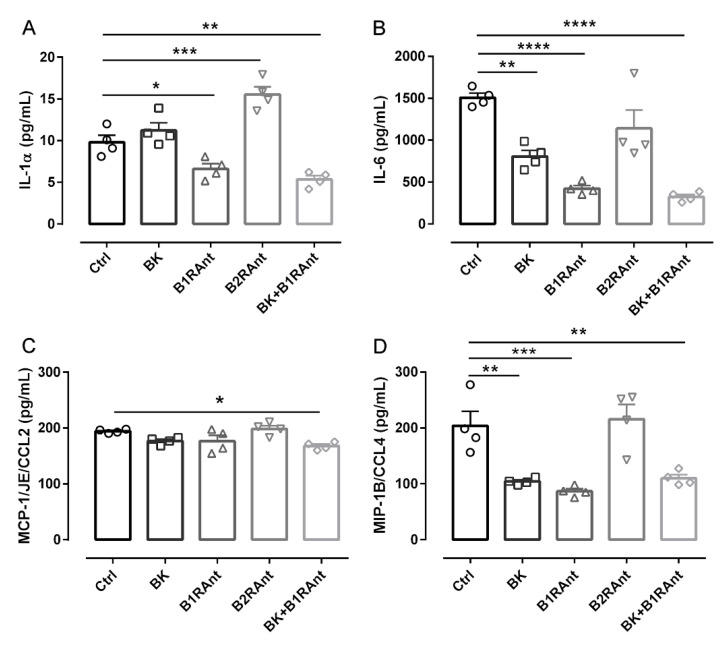
B1 and B2 receptors modulate inflammation in Alzheimer’s disease. Hippocampal slices from 12-month-old TG animals after 20 DIV of treatment with 300 pM bradykinin and/or 200 pM B1R antagonist or 300 pM B2R antagonist. The culture medium was collected and the level of cytokines, (**A**) IL-1a, (**B**) IL-6, (**C**) MCP-1/JE/CCL2 and (**D**) MIP-1B/CCL4, was evaluated using the Elisa Multiplex method with the Luminex 200 system. NT—untreated slices; BK—bradykinin 300 pM; B1RAnt—[des-Arg10]-HOE 140 200 pM; B2RAnt—Bradyzide 300 pM; * *p* < 0.05; ** *p* < 0.01; *** *p* < 0.001; **** *p* < 0.0001. Data were analyzed with one-way ANOVA followed by Dunnett’s multiple comparison test. Technical quadruplicate of biological samples of four animals were used for each treatment.

**Figure 7 pharmaceuticals-13-00288-f007:**
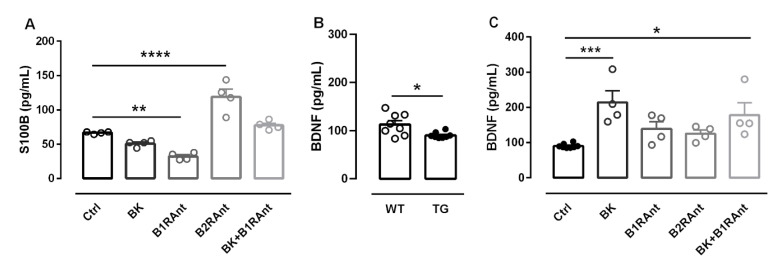
B2R is important in the process of neuronal protection provided by the KKS system. After 20 DIV of treatment with BK, B1RAnt, B2RAnt, or BK + B1RAnt, the culture medium was collected and the S100b and BDNF levels were analyzed by Elisa. (**A**) S100B concentration after treatments. (**B**) BDNF concentration in 12-month-old WT and TG animals. (**C**) BDNF levels in 12-month-old TG animals after treatments. Histograms and vertical bars show the mean ± SEM. NT—untreated slices; BK—bradykinin-treated slices; B1RAnt—slices treated with B1R antagonist [des-Arg10]-Hoe-140; B2RAnt—slices treated with B2R antagonist BDZ; BK + B1RAnt—slices treated with bradykinin and B1R antagonist [des-Arg10]-Hoe-140. * *p* < 0.05; ** *p* < 0.01; *** *p* < 0.001; **** *p* < 0.0001. (**A**,**C**) Data were analyzed with one-way ANOVA followed by Dunnett’s multiple comparison test. Data in B were analyzed with Student’s *t*-test. Biological samples of four animals were used for each treatment.

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
