# Peer review of "Kinin B2 Receptor Activation Prevents the Evolution of Alzheimer’s Disease Pathological Characteristics in a Transgenic Mouse Model"

_pharmaceuticals, 2020, doi:10.3390/ph13100288_

Round 1

Reviewer 1 Report

The investigators are studying an important topic of dementia, Alzheimer's disease, with potential treatment implication.

The Introduction should be strengthened with more background knowledge of how the KKS signaling can be modulated with potential medications that may be applied for patients. As mentioned in the Introduction and Discussion, the authors have been working on the area for some time with accumulations publications. How can their findings be translated to benefit of patients. What approved drugs can modulated the KKS system ?

For neuroinflammation, microglia and astrocytic activation should be shown by appropriate immunofluorescence or immunohistochemistry for markers of activation.

Author Response

Dear,

Thank you for your comments and contribution to improve the quality of this work. Please find below my answers, point by point, with indication of line location.

1 - The Introduction should be strengthened with more background knowledge of how the KKS signaling can be modulated with potential medications that may be applied for patients. As mentioned in the Introduction and Discussion, the authors have been working on the area for some time with accumulations publications. How can their findings be translated to benefit of patients. What approved drugs can modulated the KKS system?

Thank you for your suggestion. We believe this is introduced in the second paragraph of the Introduction section (starting at line 62). We tried to keep the introduction focused on bradykinin analogs and their receptors. It is quite difficult to explore medication acting on KKS, like ACE inhibitors, in a single sentence. The only kinin analog approved for commercial use is the B2 receptor antagonist HOE-140 (icatibant), which is used to treat hereditary angioedema. This information was added to the Introduction section. The following sentence was added to the introduction (Line 58 – 63): “In the last years, several studies suggested the inhibition or activation of B1 and B2 receptors as therapeutic targets for different brain disorders such as ischemic stroke, temporal lobe epilepsy, febrile illness, spinal cord injury, traumatic brain injury and in Alzheimer’s disease with controversial results (PMID: 29355711). However, in our knowledge, there are only two kallikrein kinin system (KKS) modulating drugs approved for commercial use, the B2 receptor antagonist HOE140 (Icatibant) and ecallantide, which are used to treat hereditary angioedema”.

Also, the following sentence was modified from “Other research from our group has shown the involvement of the kallikrein kinin system (KKS) in aging and Alzheimer's disease.” to “Other research from our group has shown the involvement of kinin receptors in aging and Alzheimer's disease.” (Line 48);

and, the following sentence was included (lines 78 and 79): “Thus, the use of B2R agonists could, therefore, be a possible therapeutic option for patients diagnosed with Alzheimer's disease.”

2 - For neuroinflammation, microglia and astrocytic activation should be shown by appropriate immunofluorescence or immunohistochemistry for markers of activation.

Thank you for your comment. Microglia and astrocytes are key mediators of neuroinflammation in neurodegenerative diseases and when activated can produce inflammatory factors. We are aware that microglia activation is an important sign of neuroinflammation but we did not succeed to obtain high-quality slices from the organotypic hippocampal cultures. This is why we show only the quantification of inflammatory molecules, which are also well accepted in the literature. Also, S100b is found, predominantly, in the cytoplasm of astrocytes, where it is actively secreted, being able to exercise both intra and extracellular functions. In the extracellular space, it can exert trophic or toxic effects, depending on its concentration. In micromolar concentrations, the protein has neurotoxic effects by inducing the release of pro-inflammatory cytokines and stimulating the release of nitric oxide by astrocytes and microglia. Several studies have shown that high concentrations of S100b lead to the activation of astrocytes and, consequently, the release of pro-inflammatory components and reactive oxygen species, thus being able to be used as a marker of neuronal damage and activation of astrocytes [Michetti, F., et al. J Neurochem, 2019. 148(2): p. 168-187, and; Wang, D.D. and A. Bordey, The astrocyte odyssey. Prog Neurobiol, 2008. 86(4): p. 342-67].

Reviewer 2 Report

This article describes the protective effects of Kinin B2 receptors in a rodent model of Alzheimer´s disease, in line with previous reports by the same group. In general, the paper is clearly written and well-structured, and provides new evidence supporting the potential therapeutic effect of kinins in neurodegeneration and neuroinflammation. The authors describe the effect of pharmacological agonists and antagonists on animal behaviour (movement, memory, blood flow...) and the pathological hallmarks of AD on brain slices of wild type and Alzheimer´s transgenic mice (protein amyloid accumulation, viability, secretion of inflammatory cytokines,...). The experimental design, structure and graphical representations of the paper are similar to recent articles by the same group (Front Aging Neurosci 2019, 11, 168, 545 doi:10.3389/fnagi.2019.00168), but the results seem to be original.

However, the paper has several formal issues that should be addressed. I attached my comments on the pdf file. Please, check the attached file and correct the issues indicated, but also revise the manuscript beyond my comments, because some issues could be repeated and remain unidentified. I summarize some of them here:

  1. All figures are multipanel, but only Fig. 7 is separated in A, B, C,... (although the text for some figures makes a reference of this division, such as for Fig.2). Multipanel figures should have each panel clearly defined and referenced in the corresponding text. If the authors are summarizing the results so much that this is separation seems unnecessary to them, then the authors are doing a poor job explaining the results and should therefore expand the results section as needed. On the other hand, Fig. 7B and 7C should be fused into one, and the same is probably true for Fig. 3.
  2. There are many issues in figure legends: internal discrepancies and missing information. In all figures: the group versus which the rest of the groups are compared or significant is never defined in the legends. The fact that some of the graphs write a line to identify them is not enough. In Fig. 2, the authors don´t state what test they used to analyze data. In Figs. 4 to 7, the authors state they used Dunnett´s test, but then in the Material and Methods section they state they used totally different statistical methods. Please, define clearly which tests you used and be consistent. There is also inconsistency in the PI concentration used in Fig. 5 and the concentration stated in the Material and Methods. They are not the same and there is a big difference between them, which could determine results.
  3. The authors abuse of abbreviations, making the reading difficult, rather than simplifying it. I am fine with the use of abbreviations in the graphs, as long as they are clearly explained in the legend and the manuscript. However, non-standard abbreviations should be avoided and have only one dimension. The authors are using abbreviations where they include both the mouse strain and the treatment altogether and use them as if it was a synonym of "animals". For example, in the sentence (1st paragraph of Results): "On day 72, TGNG291 showed a significant decrease in the running distance to reach the target compared to day 12." Where it should say: "...transgenic mice treated with NG291 showed a significant...". The same goes for the use of this multi-dimensional abbreviations in the legends or other parts of the manuscript.
  4. The discussion section is lengthy and speculative, too much for a paper with a very focused objective and result. The authors should make it considerably briefer.
  5. In the Material and Methods section, the authors specify the references of some products, but almost never specify the Country and Headquarters of the manufacturer´s, which is a basic rule in this section. References are highly appreciated, especially for products such as antibodies, for which there are many similar options within the same company and which sometimes work and sometimes not. However, the country and headquarters are traditionally more important for various reasons. Please, include them.
  6. In section 4.12 becomes clear that the authors do not state clearly how the behavioral data are represented (=what do the bars and error bars mean?), neither in the methods or the graphs or their legends, at least for Figs. 1 and 3. It is indicated only for the organotypic experiments. Furthermore, which units are they using in Figures 2A and 2B?
  7. The authors use a lot of references for such a focused paper and include almost 20% of self-citations. The total number should be put down to 50 (at least) and the self-citations significantly reduced. For example, references 43 to 47 are a lot for such a small part of the article. But please do not limit yourself to cut exactly in the parts I am stating specifically. This can be applied to most of the manuscript.
  8. In the last paragraph of the Methods section there is a confusing and most likely incorrect explanation of replication experiments. Technical replicates should not be included there, if you are using the technical replicates to calculate a mean and then use these means to calculate the mean of the biological replicates. You cannot use two times the same values to obtain a final value. Please, clarify.
  9. I don´t think the Conclusions section is really necessary
  10. In Figure 1: I am not sure a Student t-test is the right choice. You are comparing only Day 12 versus day 72 within each experimental group, but the groups are not totally independent, and they should also be compared between them. For example, WT versus TG, or CSF versus BDZ within WT or TG mice. For this you need to adjust for multiple comparison, which is what you actually state in the Material and Methods section.
  11. CSF is introduced very late as an abbreviation in the main text (the earliest is in the legend of Fig.1) and its identification is not straight-forward. Only when you read the Material and Method section far below is that you realize that it was the vehicle of the drugs. The fact that it is artificial CSF is not intuitive either and again stated very late in the text, and when you identify CSF as cerebrospinal fluid makes you wonder why would anybody extract real CSF to be the vehicle of drugs. This should be stated earlier in the Result section.
  12. A similar thing happens with the experimental design throughout the paper. It is very unclear why the authors would use antagonists against B1 and B2 receptors alone, but then only block B1 receptors in combination with BK in the organotypic slices. What was the purpose? What were they expecting to observe or prove? Why not using B2R antagonist in combination with BK too? This is barely and very briefly suggested only in the Discussion lines 297-300. The next clue is in lines 463-465 of the Material and methods section. No other indication to explain the experimental design was found by this reviewer, and it was difficult to interpret the graphs and understand the type of conclusions that could be drawn from these experiments. Please, clarify this early in the results section.
  13. Line 184: very unclear sentence. Please, clarify. There are other sentences that should be removed or simplified or clarified. Please, revise the attached file and the manuscript thoroughly before resubmission.

Author Response

Dear,

Thank you for your comments and contribution to improve the quality of this work. Please find below my answers, point by point, with indication of line location when possible, since several modifications are spread along with the text. The files contain the modification labeled in the revision mode.

1 - All figures are multipanel, but only Fig. 7 is separated in A, B, C,... (although the text for some figures makes a reference of this division, such as for Fig.2). Multipanel figures should have each panel clearly defined and referenced in the corresponding text. If the authors are summarizing the results so much that this is separation seems unnecessary to them, then the authors are doing a poor job explaining the results and should therefore expand the results section as needed. On the other hand, Fig. 7B and 7C should be fused into one, and the same is probably true for Fig. 3.

Thank you for your suggestion. We did not identify each panel with A, B, C… considering that they were clearly named. In this revised version panels were identified as A, B, C… and indicated in the text when necessary. Panels of Figure 3 were fused but we would like to keep Figure 7B and 7C separated because these individual panels have an important meaning. First, in the case of BDNF density, it is important to highlight the difference between WT and TG mice, as TG mice show a significant reduction in this factor density which means a lot to their long-term memory. As it is indicated in the text, we recently published the effects of BK and its receptor's agonist and antagonist in organotypic hippocampal culture of 12 month-old WT mice (Toricelli et al., 2020). In this way, in the present work, we did not perform this analysis again, but instead, we are showing the effects of BK and its receptor's agonist and antagonist in the organotypic hippocampal culture of TG mice. That's why it is important to keep Figures 7B and 7C separated.

2- There are many issues in figure legends: internal discrepancies and missing information. In all figures: the group versus which the rest of the groups are compared or significant is never defined in the legends. The fact that some of the graphs write a line to identify them is not enough. In Fig. 2, the authors don´t state what test they used to analyze data. In Figs. 4 to 7, the authors state they used Dunnett´s test, but then in the Material and Methods section they state they used totally different statistical methods. Please, define clearly which tests you used and be consistent. There is also inconsistency in the PI concentration used in Fig. 5 and the concentration stated in the Material and Methods. They are not the same and there is a big difference between them, which could determine results.

All the figure legends were revised and statistical tests informed. The PI concentration is 1 µg/mL and was corrected in both places. For one-way ANOVA followed by Dunnett’s multiple comparison test, values were compared to the control group (Ctrl). This is indicated in the statistical section. For one-way ANOVA followed by Tukey's test, no indication was done since this test compare all the groups, so the line will identify the different groups.    

3 -The authors abuse of abbreviations, making the reading difficult, rather than simplifying it. I am fine with the use of abbreviations in the graphs, as long as they are clearly explained in the legend and the manuscript. However, non-standard abbreviations should be avoided and have only one dimension. The authors are using abbreviations where they include both the mouse strain and the treatment altogether and use them as if it was a synonym of "animals". For example, in the sentence (1st paragraph of Results): "On day 72, TGNG291 showed a significant decrease in the running distance to reach the target compared to day 12." Where it should say: "...transgenic mice treated with NG291 showed a significant...". The same goes for the use of this multi-dimensional abbreviations in the legends or other parts of the manuscript.

Thank you. All the text was carefully revised and the number of abbreviations was reduced along with the text.  

4 -The discussion section is lengthy and speculative, too much for a paper with a very focused objective and result. The authors should make it considerably briefer.

The discussion was carefully revised and shortened as well as the number of references.

5 -In the Material and Methods section, the authors specify the references of some products, but almost never specify the Country and Headquarters of the manufacturer´s, which is a basic rule in this section. References are highly appreciated, especially for products such as antibodies, for which there are many similar options within the same company and which sometimes work and sometimes not. However, the country and headquarters are traditionally more important for various reasons. Please, include them.

Thank you. The country and headquarters were included along the methods section.

  1. In section 4.12 becomes clear that the authors do not state clearly how the behavioral data are represented (=what do the bars and error bars mean?), neither in the methods or the graphs or their legends, at least for Figs. 1 and 3. It is indicated only for the organotypic experiments. Furthermore, which units are they using in Figures 2A and 2B?

Thank you. Corrections were done in all the figures. Also, the statistical section was rewritten as follows: "Behavioral data obtained in the Barnes maze test, locomotion, rearing, and mechanosensitivity test were analyzed by one-way ANOVA, followed by Tukey’s multiple comparison tests. Data related to cerebral blood flow and figure 7B were analyzed by the Student t-test. Data related to amyloid plaques quantification and organotypic hippocampal culture were analyzed with one-way ANOVA followed by Dunnett’s multiple comparison test considering the CSF or the control group (Ctrl) the references for comparisons. All analyses were performed using GraphPad Prism 8.0 (GraphPad Software Inc., San Diego, CA, USA). All data were expressed as means ± SEM. Only probability values (p) less than 0.05 were considered statistically significant".

  1. The authors use a lot of references for such a focused paper and include almost 20% of self-citations. The total number should be put down to 50 (at least) and the self-citations significantly reduced. For example, references 43 to 47 are a lot for such a small part of the article. But please do not limit yourself to cut exactly in the parts I am stating specifically. This can be applied to most of the manuscript.

Thank you for your comments. The number of references was reduced to a total of 48. Our group has been published in this field for more than a decade and most of the references for the rationale of this paper are from our group. In this way, we think it is important to highlight the necessary information for the best comprehension of the matter. We revise our self-citations and four references were excluded.

  1. In the last paragraph of the Methods section, there is a confusing and most likely incorrect explanation of replication experiments. Technical replicates should not be included there, if you are using the technical replicates to calculate a mean and then use these means to calculate the mean of the biological replicates. You cannot use two times the same values to obtain a final value. Please, clarify.

The number of samples in the figures of organotypic cultures represents different biological samples. Technical replications were not used as “n” in the statistics. The statistical section was rewritten and this last sentence was excluded. The number of biological samples used is indicated in the legends of the figures.

  1. I don´t think the Conclusions section is really necessary

I believe this is not optional. The manuscript is prepared over a template of the journal.

  1. In Figure 1: I am not sure a Student t-test is the right choice. You are comparing only Day 12 versus day 72 within each experimental group, but the groups are not totally independent, and they should also be compared between them. For example, WT versus TG, or CSF versus BDZ within WT or TG mice. For this you need to adjust for multiple comparison, which is what you actually state in the Material and Methods section.

Thank you. Behavioral data (Figures 1 and 2) were reanalyzed according to your suggestion. In the transgenic groups, one outlier of each group was identified and removed. The final results did not change with or without them. The “Statistical Analysis” was rewritten.

  1. CSF is introduced very late as an abbreviation in the main text (the earliest is in the legend of Fig.1) and its identification is not straight-forward. Only when you read the Material and Method section far below is that you realize that it was the vehicle of the drugs. The fact that it is artificial CSF is not intuitive either and again stated very late in the text, and when you identify CSF as cerebrospinal fluid makes you wonder why would anybody extract real CSF to be the vehicle of drugs. This should be stated earlier in the Result section.

Thank you for your observation. The text was modified as follows: “Control animals were treated with artificial cerebral spinal fluid (CSF) which was the vehicle used to dilute all the peptides.” (Lines 88 and 89).

  1. A similar thing happens with the experimental design throughout the paper. It is very unclear why the authors would use antagonists against B1 and B2 receptors alone, but then only block B1 receptors in combination with BK in the organotypic slices. What was the purpose? What were they expecting to observe or prove? Why not using B2R antagonist in combination with BK too? This is barely and very briefly suggested only in the Discussion lines 297-300. The next clue is in lines 463-465 of the Material and methods section. No other indication to explain the experimental design was found by this reviewer, and it was difficult to interpret the graphs and understand the type of conclusions that could be drawn from these experiments. Please, clarify this early in the results section.

Thank you for your suggestion. The sentence explaining why to use B1 antagonist plus BK and B2 antagonist alone was inserted in the results section (Lines 178-181).

  1. Line 184: very unclear sentence. Please, clarify. There are other sentences that should be removed or simplified or clarified. Please, revise the attached file and the manuscript thoroughly before resubmission.

Thank you. The sentence was rewritten as follows:  These data show that BK acting only on the B2R, led to a drastic decrease in all evaluated cytokines. (Lines 208-209).

Round 2

Reviewer 1 Report

The study is important as it bears potential for development of novel treatment for AD.

I suggest that:

  1. immunohistochemistry/immunofluorescence experiments should be performed to document evidence of neuroinflammation by microglial activation.

    2. the authors should in their discussion propose the potential drugs that may confer effect of B2R activation.

Author Response

  1. immunohistochemistry/immunofluorescence experiments should be performed to document evidence of neuroinflammation by microglial activation.

As previously answered we did not succeed to obtain high quality slices from the organotypic hippocampal cultures. We will try to do OHC with thinner slices to perform IHC with the entire slice. We were trying that, since we want to co-localize kinin receptors with the neural cell types, but due to the pandemics our labs were closed in the beginning of March and still closed. We are restarting now the import of a new batch of transgenic mice with about 8-weeks-old.

                As we are not able to perform this experiment now and in order to address properly the referee question we remove all the reference to BKB2R decrease the neuroinflammation from the Results, Discussion, Methods and Conclusion. We report only as alterations (increase or decrease) in markers of inflammation. Please see below the modifications.

Lines 201-209

"2.6. B2R activation decreases the release of inflammatory markers.

One of the main KKS roles discussed in the literature is its effect on inflammation. To better understand the participation of B2R in the release of inflammatory markers in Alzheimer's disease, we treated the hippocampal slices with different kinin analogues.

In OHC of 12-months-old transgenic mice treatment with bradykinin significantly decreased IL-6 and CCL4 levels, two important inflammatory mediators (Figure 6B and D). Treatment with the B1R antagonist decreased the release of IL-1a, IL-6, and CCL4 (Figure 6A, B and D).

The B2R antagonist increased IL-1a levels (1.59-fold, p<0.001) (Figure 6A). These data show that BK acting only on the B2R, led to a drastic decrease in all evaluated cytokines."

Lines 300-310. The last sentence of this paragraph states that the present work has the limitation to do not show microglia activation by immunohistochemical.

"Interestingly, BK treatment together with the BKB1R antagonist was able to significantly decrease the levels of all the inflammatory mediators analyzed. However, the role of the BKB1R cannot be discarded; Sanden et al. demonstrated that BKB1R is upregulated after lipopolysaccharide (LPS) injection, resulting in an increase in TNFα and IFNɣ [33]. Another study recently published by our group demonstrated the importance of the animals' age in respect of the neuroprotective action of bradykinin and BKB2R [12]. In this study, a decrease in inflammatory markers was observed after treatment with BK only in 12-month-old animals; in 6-month-old animals the kinin receptor activation increased inflammatory markers [12]. Due to technical limitations in obtain acceptable thinner slices from organotypic hippocampal culture the present work has the limitation to do not show microglia activation by immunohistochemical."

Lines 489-497.

"4.11. Inflammatory Markers Evaluation

Inflammatory markers were evaluated by assessing the levels of MCP-1/JE/CCL2, MIP- 1B/CCL4, interleukin 1α and 6. These markers were measured in the OHCs culture medium using a Magnetic Luminex® Assay—Mouse Premixed Multi-Analyte Kit R&D systems (Minneapolis, Minnesota, USA), according to the manufacturer’s instruction.

Release of BDNF and S100b by OHCs were quantified in the medium by Quantikine Elisa Total BDNF Immunoassay® (R&D systems, Minneapolis, Minnesota, USA) and by Mouse S100b (S100 Calcium Binding Protein B) Elisa Kit (Elabscience, Houston, USA), respectively, following the manufacturer’s instructions."

Line 508-516.

"5. Conclusions

In this study, the importance of the BKB2R in respect of spatial memory improvement, a reduction of amyloid plaques, and increased blood flow in transgenic Alzheimer's disease mice was shown. Moreover, in 12-TgOHC we observed that the modulation of the kallikrein kinin system was important to decrease cell death and cell damage, Inflammatory markers and increase in BDNF levels. These data allow us to highlight BKB2R and BKB1R and the possible important role they might play in pharmacotherapy for Alzheimer’s disease. In this way, the present study is one step further in the comprehension of the involvement and importance of the kallikrein-kinin system in Alzheimer’s disease and its pharmacological modulation."

  1. the authors should in their discussion propose the potential drugs that may confer effect of B2R activation.

As suggested, a paragraph was included in the Discussion Section (Lines 349-356), as below:

" Therefore, in the present study, we demonstrated that the activation of BKB2R by the specific agonists NG291 and BK can become an important tool in the fight against Alzheimer's disease. Other known drugs can also activate B2R and could be potential tools for the treatment of neurodegenerative diseases, but their therapeutic value, as well as side effects considering the treatment of the elderly, are still to be determinate. In this way, the angiotensin-converting enzyme inhibitor (ACEI) can increase the concentration of circulating BK and increase B2R functions as an allosteric enhancer potentiating the actions of BK. Also, there is the angiotensin II receptor blocker Losartan, which acts as a partial agonist of B2R."